# CXCR1 Expression in MDA-PCa-2b Cell Upregulates ITM2A to Inhibit Tumor Growth

**DOI:** 10.3390/cancers16244138

**Published:** 2024-12-11

**Authors:** Timothy O. Adekoya, Nikia Smith, Parag Kothari, Monique A. Dacanay, Yahui Li, Ricardo M. Richardson

**Affiliations:** 1Cancer Research Program, Julius L. Chambers Biomedical/Biotechnology Research Institute, North Carolina Central University, Durham, NC 27707, USA; 2Department of Biological & Biomedical Sciences, North Carolina Central University, Durham, NC 27707, USA

**Keywords:** interleukin-8 receptors, CXCR1, CXCR2, ITM2A, prostate cancer, androgen receptor, prostate-specific antigen

## Abstract

Prostate cancer constitutes a large percentage of cancer-related deaths among men. Due to the complexity of the disease process and resistance to androgen deprivation therapy, a lot remains unknown about prostate tumorigenesis. Recently, several chemokines, including CXCL8 and its receptors (CXCR1 and CXCR2), have been shown to impact the development and progression of prostate cancer. In this study, we used various assays and animal models to better understand the role of CXCL8 receptors in prostate cancer incidence. The results herein indicate that the receptors may have opposing effects in the onset of prostate cancer. CXCR1 expression, not CXCR2, upregulates the tumor suppressor ITM2A to inhibit prostate tumor growth.

## 1. Introduction

Prostate cancer (PCa) is the most diagnosed cancer type affecting the male population, with over 240,000 new cases (representing about 26% of all new cancer cases) reported in the USA alone in 2021 [1]. While there has been a gradual decline in the overall incidence and mortality of PCa in recent years owing to early detection and treatment successes, the development of the metastatic form of the disease in patients has remained a continuous source of clinical/therapeutic challenge [2]. The cells of the prostate depend on androgen and its receptor, androgen receptor (AR), to maintain normal homeostasis and development [3,4]. It is, therefore, not surprising that aberrant expression, activation, gene mutations, and the dysregulation of biomolecules along the androgen–AR pathway occur in prostate cancer tissues [5]. Consequently, the AR signaling pathway along with its regulatory mechanisms has been a target for treatment and an area of extensive biological studies over the years. With AR-targeted therapies, patients experience tumor regression/remission. However, prolonged ADT ultimately results in the development of castration-resistant PCa, an often aggressive metastatic form of PCa [6,7]. In addition, several studies have revealed the involvement of various androgen-independent pathways in the promotion of prostate tumorigenesis [8,9].

CXCR1 and CXCR2 are members of the CXC subfamily of chemokine receptors that interact with interleukin-8 (IL-8/CXCL8) to mediate signal transduction [10,11]. Although they exhibit high amino acid sequence homology, their selectivity and affinity to couple with CXC chemokines greatly differs. In humans, CXCR2 can be activated by CXCL1-3 and CXCL5-8; whereas, CXCR1 is activated by CXCL6 and CXCL8 [12,13]. CXCR1 and CXCR2 are expressed in various cell types including immune cells, fibroblasts, endothelial cells, osteoclasts, and cancer cells [14,15,16]. Apart from helping to maintain homeostasis and physiological communication, both CXCR1 and CXCR2 have been shown to be aberrantly expressed or activated in various cancer types, including PCa [17,18,19,20]. The role of CXCR2 in PCa development and progression has been assessed in several studies [21,22,23]. To date, however, limited information exists with regard to the role of CXCR1 in PCa development and progression.

In this study, we investigated the role of CXCR1 expression in prostate tumor, using the androgen-dependent cell line MDA-PCa-2b to stably express CXCR1 (MDA-PCa-2b-CXCR1) and, for control, cells expressing CXCR2 (MDA-PCa-2b-CXCR2) or vector alone (MDA-PCa-2b-vec). CXCR1 expression, not CXCR2, was shown to upregulate the tumor suppressor ITM2A to diminish cell proliferation and inhibit tumor growth. Since CXCR2 expression was shown to promote PCa growth [21,22,23] the data likely indicate that CXCR1 and CXCR2 play opposing roles in prostate tumorigenesis.

## 2. Materials and Methods

### 2.1. Materials

The PCa cell lines MDA-PCa-2b (CRL-2422) and LNCaP (CRL-1740) were purchased from American Type Culture Collection (ATCC) (Manassas, VA, USA). These cells were cytogenetically tested and authenticated via the short tandem repeat method prior to being frozen. All cell-based experiments were carried out using only cells that had been tested and cultured for less than 12 weeks. Cell culture and molecular biology reagents, including fetal bovine serum (FBS) (cat # 16140-071), RPMI media (cat # 11875-093), and Dulbecco’s modified Eagle’s medium (DMEM/Ham’s F12) (cat # 10565-018) were sourced from Life Technologies (Carlsbad, CA, USA). Hydrocortisone (H6909), epidermal growth factor (EGF) (E4127), human insulin (I9298), cholera toxin (C8052), phosphoethanolamine (P0503), protease inhibitors cocktail (P8340), selenious acid (211176), and radioimmunoprecipitation assay (RIPA) buffer (R0278) were purchased from Sigma-Aldrich (St. Louis, MO, USA). XTT Cell Proliferation Assay Kit (30-1011K) was purchased from ATCC, CXCL8 (200-08M) from Peprotech (East Windsor, NJ, USA), PE-conjugated anti-mouse IgG (115-116-146) from Jackson ImmunoResearch, and Geneticin (G418) (10131-027) from Life Technologies, while UltraPure LMP agarose (16520-050) and Lipofectamine 3000 (L3000-015) were purchased from Invitrogen (Waltham, MA, USA). The proteome profiler human chemokine array kit (ARY017) and proteome profiler mouse chemokine array kit (ARY020) were sourced from R&D systems (Minneapolis, MN, USA). Human CXCR1 cDNA (OHu19357D), human CXCR2 cDNA (OHu23649D), human ITM2A (OHu18395D), all in pcDNA3.1-C-(k)DYK plasmid backbone, as well as control pcDNA3.1-C-(k)DYK plasmid, were obtained from GenScript (Piscataway, NJ, USA). Nude (Nu/J) mice were purchased from Jackson Laboratory (Bar Harbor, ME, USA). All other reagents were obtained from commercial sources.

### 2.2. Cell Culture

MDA-PCa-2b cells were cultured in DMEM/Ham’s F12 media that was supplemented with 25 ng/mL cholera toxin, 20% heat-inactivated FBS, 10 ng/mL EGF, 45 nM selenious acid, 0.005 mM phosphoethanolamine, 100 pg/mL hydrocortisone, 0.005 mg/mL human insulin, 100 µg/mL streptomycin, and 100 IU/mL penicillin. LNCaP cells were cultured in RPMI 1640 media supplemented with 10% FBS, 1.26 g/L glucose, 1 mM sodium pyruvate, 10 mM HEPES buffer, 100 µg/mL penicillin, and 100 IU/mL streptomycin. Cell harvesting was performed using 0.05% trypsin-EDTA (1X). All cells were maintained at 37 °C and with 5% CO_2_ air atmosphere.

### 2.3. Generation of Cells Overexpressing Receptors (CXCR1, CXCR2), ITM2A, and GFP-Positive in PCa Cell Lines

For the overexpression of CXCR1 and CXCR2 in MDA-PCa-2b and LNCaP cells, cDNA constructs of these receptors in pcDNA3.1-C-(k)DYK plasmid obtained from GenScript were utilized. Briefly, 4 × 10^6^ cells were plated for 48 h in 100 mm^3^ dishes. Cells were transfected with 40 µg plasmid containing either CXCR1, CXCR2, or empty vector using Lipofectamine 3000, as described by the manufacturer instructions. Twenty-four hours post-transfection, cells were incubated with G418 (1 mg/mL for MDA-PCa-2b and 400 µg/mL for LNCaP) for four weeks. Geneticin-resistant cells were sorted by FACS using CXCR1 or CXCR2 antibodies.

For ITM2A overexpression in MDA-PCa-2b, 4 × 10^6^ cells were plated for 48 h in 100 mm^3^ dishes, following which, Lipofectamine 3000 was utilized to transfect 40 µg human ITM2A plasmid or control vector into cells. Transfected cells were subsequently incubated with G418 (1 mg/mL) for 4 weeks and geneticin-resistant single clones were isolated, screened, and subsequently utilized for further assays.

For the generation of GFP-positive PCa cells, pcDNA3.1-GFP plasmids were transfected with either MDA-PCa-2b-Vec, MDA-PCa-2b-CXCR1, or MDA-PCa-2b-CXCR2 cells. Briefly, 2 × 10^6^ cells were plated for 48 h in 6-well plates, following which, Lipofectamine 3000 was utilized to transfect 10 µg pcDNA3.1-GFP plasmids into respective cells. Forty-eight hours post-transfection, the morphology of GFP-positive cells was subsequently visualized using a Keyence BZ-X810 fluorescent microscope (Keyence, Itasca, IL, USA).

### 2.4. Cell Proliferation Assay

Cell proliferation was determined using 3-(4,5-dimethylthiazol-2-yl)-2,5-diphenyltetrazolium bromide (MTT) reagent (MP Biochemicals, Santa Ana, CA, USA; cat # 102227). Briefly, adherent cells were harvested by use of 1X trypsin and detached cells collected by centrifugation. Cells (5 × 10^3^) were subsequently seeded in 100 μL of growth media in each well of a 96-well plate and incubated for different time points. At 48, 96, and 144 h, 5 µL of MTT solution (5 mg/mL) was added to cells in each well, and the plate was incubated in a 5% CO_2_ incubator at 37 °C for 2 h. MTT solution was subsequently aspirated from the wells, and 200 µL dimethyl sulfoxide (DMSO) was added to dissolve the produced formazan crystals. Using a SpectraMax 190 microplate reader (Molecular Devices, San Jose, CA, USA), the absorbance reading was measured at 570 nm. Blank wells were used for background control.

### 2.5. Apoptosis Assay

For apoptosis, CellEvent^TM^ caspase-3/7 green ready probe reagent (R37111) and NucRed^TM^ live 647 ready probe reagent (R37106) (ThermoFisher Scientific, Waltham, MA, USA) were used. Briefly, cells (1 × 10^4^) were seeded in 100 μL of culture media per well in a 96-well plate (Corning, Cat #3603) and incubated overnight. Next, growth media was removed from cells and replaced with media containing 2 drops/mL of both CellEvent^TM^ caspase-3/7 green and NucRed^TM^ live 647 reagents. After 30 min of exposure to reagents, images of apoptotic cells (stained by the caspase-3/7 probe) and total live cells (stained by the NucRed^TM^ live 647 probe) were captured using IncuCyte ZOOM^®^ live-cell system (Essen Bioscience, Ann Arbor, MI, USA). Obtained results were subsequently analyzed using the IncuCyte S3 software and the data presented as an apoptotic index ([apoptotic cells/total cells] × 100).

### 2.6. Chemotaxis

Chemotaxis was assessed using 96-well ChemoTX system plates (ChemoTx 101-8) commercially obtained from Neuro Probe Inc (Gaithersburg, MD, USA). Briefly, 30 µL of growth media containing different concentration of IL-8 were initially placed in the bottom wells of the chemotaxis plate, following which, cells (1 × 10^4^) were suspended in 25 µL media placed on each spot of the ChemoTX filter. Migration was allowed to continue for 24 h at 37 °C in humidified air containing 5% CO_2_. The filter membrane was subsequently removed and migrated cells in wells counted. The results are represented as a chemotaxis index (mean number of cells for chemokine dilution/mean number of cells for media alone). The results are representative of two separate experiments.

### 2.7. Clonogenic Assay

Clonogenic analysis was carried out as described by Franken et al. [24]. Briefly, colony formation assay was conducted by seeding 2.5 × 10^3^ cells (MDA-PCa-2b-Vec, MDA-PCa-2b-CXCR1, and MDA-PCa-2b-CXCR2) in a 6-well plate, using 2 mL media. Cells were then incubated, and colony formation was monitored over 3 weeks. The images of single cell colonies were subsequently obtained using a Moticam imaging capture (San Antonio, TX, USA), mounted on a microscope.

### 2.8. RNA Sequencing and Data Analysis

Total RNA in the cells was extracted using the Qiagen RNeasy kit, per the manufacturer’s protocol. Library construction, quality control, and high throughput Illumina sequencing were conducted by Novogene Corporation Inc (Sacramento, CA, USA). Briefly, raw data were processed through fastp software, read number counts ascertained through featureCounts v1.5.0-p3, and differential expression analyzed using DESeq2 R package (1.20.0). Gene Set Enrichment Analysis (GSEA) was completed, and statistical significance was assigned with an adjusted *p*-value < 0.05.

### 2.9. Gene Expression Database Analysis

The UALCAN (http://ualcan.path.uab.edu/ (accessed on 24 May 2024) data extraction and analysis platform contains TCGA (The Cancer Genome Atlas) database, along with an analysis interface [25]. We, therefore, used UALCAN to extract and evaluate the transcript level (per million) of itm2a between normal prostate and prostate adenocarcinoma tissue. Our mining parameters are “Gene: itm2a; TCGA dataset: prostate adenocarcinoma”.

### 2.10. Tumor Xenografts

All animal studies and procedures were approved by and conformed to the guidelines of the Animal Care and Use Committee of North Carolina Central University, Durham, NC, USA. Both male athymic nude mice as well as NOD scid gamma (NSG) mice are commercially available and were obtained from Jackson Laboratory. For tumor xenografts, 6–8 weeks old male athymic nude or NSG mice were injected subcutaneously with 5 × 10^6^ cells of either MDA-PCa-2b-CXCR1, MDA-PCa-2b-CXCR2, or control cells, then resuspended in a 300 μL mix of growth media and Matrigel (1.5:1 ratio). Tumor progression was then monitored and measured over the course of the experiment. After 5–10 weeks (depending on the cell type), mice were euthanized by CO_2_ asphyxiation. Tumors were harvested from mice, then weighed and measured using a Thorpe caliper. Tumor volume was calculated using the formula: Volume = (d_1_ × d_2_ × d_3_) × 0.5236; wherein d_n_ represents the three orthogonal diameter measurements [26]. Harvested tumors were frozen at −80 °C and subsequently used for further experiments.

### 2.11. Chemokine Measurement in Tumor

Chemokine levels were measured using the commercially available human (ARY017) and mouse (ARY020) chemokine array (R&D Systems), according to the protocol supplied by the manufacturer. Briefly, tumor lysates were prepared by mincing harvested tumors and lysing them in RIPA; they were then assayed for protein concentrations. Four hundred and fifty micrograms of total protein were assayed on each array, and the blots subsequently imaged and respective chemokine dots quantified for each sample.

### 2.12. FACS Analysis of Single Cell Isolates from Tumor Xenografts

Tumor xenografts were harvested from mice (n = 4), minced to a fine slurry using a dissecting blade, and incubated in digestion buffer (RPMI 1640, 5% FBS, 30 µg/mL DNAse, and 1 mg/mL collagenase) at 37 °C for 45 min. Filtered single cells were washed, cells counted, and their viability determined using trypan blue exclusion on a hemocytometer. Cells (2 × 10^6^) were resuspended in FACS analysis buffer and stained with PE-conjugated anti-mouse Ly6G (Cat #551461), CD45 (Cat #553081), or NKp46 (Cat #560757) antibodies from BD Biosciences (San Jose, CA, USA). Stained cells were analyzed on a FACScan Flow Cytometer using CellQuest software version 5.1. Unstained cells were used as negative controls and for establishing analysis gating.

### 2.13. Immunoblotting

Mouse anti-human CXCR1 (555937) and CXCR2 (555932) antibodies were obtained from BD-Pharmingen (San Diego, CA, USA). GAPDH (14C10), E-cadherin (24E10), N-cadherin (D4R1H), vimentin (D21H3), PSA (D2A8), S473 phospho-AKT (D9E), and AKT (9272S) antibodies were obtained from Cell Signaling Technologies (Danvers, MA, USA). anti-AR (ab133273), anti-VEGF (ab46154), anti-ITM2A (ab279387), and anti-TUSC3 (ab230520) were purchased from Abcam (Cambridge, MA, USA) while anti-TUSC3 (SAB4503183), anti-mouse IgG (A9044), and anti-rabbit IgG (A9169) were obtained from Sigma-Aldrich.

For immunoblotting, collected cells or minced tissues were washed and lysed in RIPA buffer that was supplemented with protease inhibitor cocktail. Cell or tissue lysates were cleared by centrifugation for 10 min, amounts of protein in each sample estimated, and 20–50 µg total protein suspended in 4× loading buffer. Samples were subsequently heated at 90 °C for 5 min and separated by SDS-PAGE. Resolved proteins were then transferred onto nitrocellulose paper and probed by using antibodies against the different proteins as indicated.

### 2.14. Statistical Analysis

The results are expressed as mean ± SEM. Statistical analysis was performed using the GraphPad Prism 6.0 (GraphPad Software, San Diego, CA, USA). Differences between the experimental groups were determined using either one-way ANOVA or Student *t* test (two-tailed), as appropriate. A *p* value < 0.05 was considered statistically significant.

## 3. Results

### 3.1. Generation and Characterization of MDA-PCa-2b Cells Stably Expressing CXCR1 and CXCR2

To determine the effect of CXCR1 and CXCR2 overexpression in MDA-PCa-2b cells tumorigenesis, we generated MDA-PCa-2b cell lines, stably overexpressing human CXCR1 (MDA-PCa-2b-CXCR1) or CXCR2 (MDA-PCa-2b-CXCR2), as well as control cells expressing the vector alone (MDA-PCa-2b-Vec). All three cell lines were assayed for receptor expression using CXCR1- and CXCR2-specific antibodies. Figure 1A depicts the flow cytometry analysis of the cell surface expression of CXCR1 (middle panel) and CXCR2 (right panel), relative to the control cells (left panel). To confirm receptor expression, cell lysates from MDA-PCa-2b-CXCR1, MDA-PCa2b-CXCR2, and MDA-PCa-2b-Vec were assayed by Western blotting. As shown in Figure 1B, CXCR1 (top panel, lane 2) and CXCR2 (middle panel, lane 3) expression were detected in MDA-PCa-2b-CXCR1 and MDA-PCa2b-CXCR2, respectively. No receptors were detected in control MDAPCa-2b-Vec cells (lane 1).

Clonogenic assay with GFP-transfected cells showed that MDA-PCa-2b-CXCR1 cells are more spread out, as compared to MDA-PCa-2b-CXCR2 and MDA-PCa-2b-Vec, which displayed a more 3-dimensional-like morphology (Figure 1C, upper panel). MDA-PCa-2b-CXCR1 cells (Figure 1C, lower panel) also showed a more spindle-like phenotype relative to MDA-PCa-2b-CXCR2 and control cells, which displayed a more cuboidal-like phenotype.

We next measured receptor responses to CXCL8 by monitoring for CXCL8-mediated chemotaxis and AKT phosphorylation. As shown in Figure 1D, both MDA-PCa-2b-CXCR1 and MDA-PCa-2b-CXCR2 cells induced dose-dependent chemotaxis in response to CXCL8. The response, however, was significantly greater in CXCR2-expressing cells, as compared to CXCR1. CXCL8 also induced the time-dependent phosphorylation of AKT in both MDA-PCa-2b-CXCR1 and MDA-PCa-2b-CXCR2 (Figure 1E). AKT phosphorylation in MDA-PCa-2b-CXCR1, however, was markedly downregulated, as compared to MDA-PCa-2b-CXCR2 and control cells (Figure 1E).

### 3.2. Effects of CXCR1 and CXCR2 Overexpression on AR, PSA, and Epithelial–Mesenchymal Transition Markers

It was previously shown that the overexpression of CXCR2 in LNCaP cells decreased AR and PSA expression. This was attributed to a switch of the cells from androgen-dependent to the more aggressive androgen-independent phenotype by CXCR2 [23]. We next determined the effect of CXCR1 overexpression in AR and PSA expression. MDA-PCa-2b-CXCR2 cells (Figure 2A, upper panel) exhibited a significant decrease in AR expression (0.651 ± 0.107, Figure 2B) relative to MDA-PCa-2b-CXCR1 (1.099 ± 0.069) and MDA-PCa-2b-Vec (1.0 ± 0.013) cells (Figure 2B).

PSA expression, in contrast, was decreased in MDA-PCa-2b-CXCR1 (0.701 ± 0.041, Figure 2A, lower panel) relative to MDA-PCa-2b-Vec (1.000 ± 0.005) cells. No significant effect was shown in MDA-PCa-2b-CXCR2 (0.986 ± 0.014) cells (Figure 2C).

Epithelial–mesenchymal transition (EMT) has been described as indicative of enhanced stemness and metastatic potential in PCa [27,28,29]. To assess the effect of CXCR1 and CXCR2 overexpression on EMT markers, the cells were assayed for E-cadherin, N-cadherin, and vimentin expression. As shown in Figure 2D, MDA-PCa-2b-CXCR1 cells displayed a significant decrease in E-cadherin (0.764 ± 0.038; Figure 2D,E) but a marked increase in N-cadherin (1.943 ± 0.292; Figure 2D,F) and vimentin (2.689 ± 0.333; Figure 2D,G) expression, relative to control MDA-PCa-2b-Vec cells (E-cad: 1.0 ± 0.02; N-cad: 1.0 ± 0.011; and vimentin: 1.0 ± 0.001). The overexpression of CXCR2 had no significant effect on E-cadherin (0.849 ± 0.036), N-cadherin (0.545 ± 0.099), or vimentin (1.301 ± 0.144) expression in MDA-PCa-2b cells, when compared to control MDA-PCa-2b-Vec cells (E-cad: 1.0 ± 0.02; N-cad: 1.0 ± 0.011; and vimentin: 1.0 ± 0.001), respectively (Figure 2D–G).

### 3.3. Effect of CXCR1 and CXCR2 Overexpression in Cell Proliferation and Tumor Growth

We next evaluated the effect of CXCR1 and CXCR2 overexpression in cell proliferation using the MTT assay. As shown in Figure 3A, MDA-PCa-2b-CXCR2 (1.030 ± 0.037) proliferated at a faster rate than MDA-PCa-2b-Vec (0.863 ± 0.011). MDA-PCa-2b-CXCR1 (0.709 ± 0.045), however, showed a significant decrease in cell proliferation relative to MDA-PCa-2b-Vec (0.863 ± 0.011) (Figure 3A). MDA-PCa-2b-CXCR1 also displayed a greater, but not significant, apoptotic index relative to control MDA-PCa-2b-Vec and MDA-PCa-2b-CXCR2 cells (Appendix A).

Xenografts in nude mice were carried out to evaluate in vivo the effect of CXCR1 and CXCR2 receptor overexpression in MDA-PCa-2b tumorigenesis. Nude mice of 6–8 weeks of age were injected with (~5 × 10^6^) either MDA-PCa-2b-CXCR1, MDA-PCa-2b-CXCR2, or control cells. Heterotopic tumor growth in animals was monitored for 10 weeks following the subcutaneous injection of cells. As shown in Figure 3, MDA-PCa-2b-CXCR1 cells developed smaller tumors (0.066 ± 0.011g), as compared to control MDA-PCa-2b-Vec (0.830 ± 0.171 g). MDA-PCa-2b-CXCR2 (1.507 ± 0.271 g), however, developed significantly bigger tumors than control MDA-PCa-2b-Vec (0.830 ± 0.171 g) (Figure 3B–D).

The overexpression of CXCR2 in LNCaP cells was previously shown to increase tumor growth [23]. To validate the results obtained with MDA-PCa-2b cells, we stably overexpressed CXCR1 (LNCaP-CXCR1) and CXCR2 (LNCaP-CXCR2) in LNCaP cells. The cell surface expression of the receptors was also confirmed by both flow cytometry (Figure 4A) and Western blot analysis (Figure 4B). As was the case for MDA-PCa-2b, the CXCL8-induced, time-dependent phosphorylation of AKT was significantly higher in LNCaP-CXCR2, when compared to LNCaP-CXCR1 and LNCaP-Vec cells (Figure 4C). Similarly, cell proliferation was increased in LNCaP-CXCR2 but decreased in LNCaP-CXCR1 cells relative to control LNCaP-Vec cells (Figure 4D). LNCaP-CXCR2 cells also showed a significant decrease in AR (0.753 ± 0.108) expression relative to LNCaP-CXCR1 (0.938 ± 0.159) and LNCaP-Vec (1.008 ± 0.008) cells (Appendix A). As for MDA-PCa-2b-CXCR1, LNCaP-CXCR1 also exhibited a very significant decrease in PSA (0.234 ± 0.077) expression relative to LNCaP-CXCR2 (0.797 ± 0.081) and LNCaP-Vec (0.986 ± 0.014) cells (Appendix A).

We also determined the effect of CXCR1 and CXCR2 overexpression in tumor growth in nude mice xenografts. As was previously shown [23], LNCaP-CXCR2 cells (1.221 ± 0.138 g) developed significantly bigger tumors relative to control LNCaP-Vec cells (0.727 ± 0.180 g) (Figure 4E). Tumors from LNCaP-CXCR1 cells (0.318 ± 0.123 g), however, were significantly smaller than LNCaP-Vec (0.727 ± 0.180 g) cells (Figure 4E,F). These results mirrored the ones obtained with MDA-PCa-2b cells and further suggest that CXCR1 and CXCR2 expression may have opposite effects in prostate tumorigenesis.

### 3.4. Effects of CXCR1 and CXCR2 in Chemokines and VEGF Expression and Leukocytes Accumulation in the Tumor Microenvironment

Chemokines are known to mediate a variety of functions in prostate tumorigenesis including angiogenesis, tumor progression, and metastasis [2,30,31]. To further examine the opposite effects of CXCR1 and CXCR2 overexpression in PCa development, we assessed the intratumoral chemokine expression in the xenograft models using mouse (Figure 5A) proteome profiler chemokine array kits. As shown in Figure 5B and Appendix A, mouse chemokines CCL21 (~224-fold), CXCL13 (~66-fold), CCL6 (~17-fold), CCL11 (~18-fold), CCL8 (~3-fold), CCL12 (~4-fold), CCL9/10 (~9-fold), CCL5 (~32-fold), and CXCL12 (~6-fold) were significantly upregulated in MDA-PCa-2b-CXCR1, as compared to MDA-PCa-2b-CXCR2 and MDA-PCa-2b-Vec tumors. Additionally, interleukin-16 (IL-16, ~10-fold), chemerin (~5-fold), and complement factor D (~3-fold) were also upregulated in MDA-PCa-2b-CXCR1 tumors.

We next analyzed the intratumor leukocyte infiltration of cell isolates from tumors. As shown in Figure 5C, both MDA-PCa-2b-CXCR1 (5.075 ± 1.284) and MDA-PCa-2b-CXCR2 (5.175 ± 1.162) showed a significant reduction in tumor infiltrating CD45^+^ cells relative to MDA-PCa-2b-Vec (8.150 ± 1.471). PMN infiltration was also reduced significantly in MDA-PCa-2b-CXCR1 (2.350 ± 0.412) tumors but not in MDA-PCa-2b-CXCR2 (6.700 ± 4.825), as compared MDA-PCa-2b-Vec (9.775 ± 2.758) tumors (Figure 5D). Interestingly, MDA-PCa-2b-CXCR1 (2.5 ± 1.675) tumors showed a ~5-fold increase in NK cell accumulation in the TME versus ~2.5-fold in MDA-PCa-2b-CXCR2 (1.600 ± 0.887), relative to control MDA-PCa-2b-Vec (0.55 ± 0.129) tumors (Figure 5E).

VEGF overexpression in tumors has been shown to correlate with increased angiogenesis, proliferation, and metastasis [32]. We next determine whether the inhibition of tumor growth in CXCR1-expressing cells correlated with a decrease in VEGF expression. As shown in Figure 5F, tumor lysates from MDA-PCa-2b-CXCR1 (0.618 ± 0.013) xenografts displayed a ~40% decrease in VEGF expression relative to MDA-PCa-2b-CXCR2 (0.997 ± 0.110) and MDA-PCa-2b-Vec (1.000 ± 0.074).

Using the NOD scid gamma (NSG) mouse model, we subsequently assessed whether the increased population of intratumor NK cells observed in the MDA-PCa-2b-CXCR1 xenografts was the major contributing factor for the marked inhibition of its tumor growth. These NSG mice lack functional B and T cells, as well as NK cell populations, due to genetic alterations in their il2rg and Prkdc genes. As shown in Figure 5G–I, the results mirrored the ones obtained with nude mice. Tumors growth was suppressed in MDA-PCa-2b-CXCR1 (0.070 ± 0.008 g) but promoted in MDA-PCa-2b-CXCR2 (2.272 ± 0.684 g) cells relative to MDA-PCa-2b-Vec cells (1.644 ± 0.617 g). These data suggest that the suppressive effect of CXCR1 on MDA-PCa-2b tumorigenesis is independent of NK cell accumulation on the TME.

### 3.5. CXCR1 Expression Upregulates the Tumor Suppressor ITM2A

To further investigate the effect of CXCR1 expression in tumor growth, we carried out RNA sequencing analysis of MDA-PCa-2b-Vec, MDA-PCa-2b-CXCR1, and MDA-PCa-2b-CXCR2 cells. Figure 6A shows a heatmap of the differential expression of genes in the generated cells. Volcano plot analysis revealed the significant downregulation of 432 genes and the upregulation of 327 genes in MDA-PCa-2b-CXCR2 cells relative to MDA-PCa-2b-Vec cells (Figure 6B). The analysis of MDA-PCa-2b-CXCR1 cells showed the downregulation of 737 genes and the upregulation of 754 genes, relative to MDA-PCa-2b-Vec (Figure 6C). Among the top 10 upregulated genes in MDA-PCa-2b-CXCR1 cells featured two tumor suppressors, ITM2A and TUSC3 (tumor suppressor candidate 3). Gene set enrichment analysis (GSEA) of the RNA sequencing data further revealed the downregulation of genes associated with cancer pathways [NES = −1.587; FDR = 0.22] in MDA-PCa-2b-CXCR1 cells (Figure 6D).

We subsequently analyzed MDA-PCa-2b-CXCR1 cells for ITM2A and TUSC3 expression. Significant upregulation was detected for only ITM2A (about 2-fold over control cells, Figure 6E) but not for TUCS3. Analysis of the TCGA dataset, using the UALCAN analysis platform [25], also revealed a significant downregulation of ITM2A gene transcripts in human prostate adenocarcinoma patients compared to control subjects (Figure 6F).

### 3.6. ITM2A Overexpression Suppresses In Vitro and In Vivo Growth of MDA-PCa-2b Cells

To determine the effect of ITM2A in MDA-PCa-2b-CXCR1 tumor growth, we first attempted to generate MDA-PCa-2b cell lines deficient in ITM2A expression. Our approaches to suppress ITM2A expression in MDA-PCa-2b-CXCR1 cells using both CRISPR (four targets; Santa Cruz, Origene, and GeneScript) and siRNA (two targets, GeneScript) failed. To overcome this problem, we generated MDA-PCa-2b cell lines stably overexpressing human ITM2A (MDA-PCa-2b-ITM2A-GFP) and control expressing the vector (MDA-PCa-2b-eGFP). ITM2A overexpression was confirmed by Western blotting (Figure 7A). MDA-PCa-2b-ITM2A-GFP cells (0.854 ± 0.445 g) showed a marked decrease in tumor growth in nude mice, as compared to control MDA-PCa-2b-GFP cells (2.066 ± 0.336 g) (Figure 7B–D).

## 4. Discussion

CXCL8 is overexpressed in the serum of PCa patients and is believed to play a key role in the pathogenesis of the androgen-independent growth of PCa cells [33,34,35]. The receptors for CXCL8, CXCR1, and CXCR2 are also overexpressed in malignant PCa but in different cells [17,20]. CXCR2 is overexpressed predominantly in neuroendocrine tumor (NE) cells, which produce CXCL8; whereas, CXCR1 expression appears to be restricted to non-NE tumor cells [20]. CXCR2 was also shown to stimulate the secretion of proangiogenic factors and promote angiogenesis, as well as metastasis [11,17,36,37,38]. To date, the role of CXCR1 expression in PCa tumorigenesis remains ill-defined. In this study, we overexpressed CXCR1 and CXCR2 in the androgen-dependent MDA-PCa-2b to determine the contribution of CXCR1 in PCa development and progression. The data herein revealed an opposite effect of CXCR1 relative to CXCR2 in the onset of PCa. CXCR1 expression in both MDA-PCa-2b and LNCaP cells inhibited tumor growth; whereas, CXCR2 expression, as previously reported, promoted tumor growth (Figure 3 and Figure 4).

The transition from androgen-dependent to androgen-independent is considered a key factor in PCa progression and resistance to hormonal therapy [11,36,37]. CXCL8 was previously shown to promote androgen-independent growth and the migration of PCa cells [35,39,40]. Li et al. [23] recently reported that CXCR2 expression was necessary to induce therapeutic resistance in PCa. In this study, CXCR2 expression, but not CXCR1, was also shown to downregulate AR expression in both MDA-PCa-2b (Figure 2A,B) and LNCaP. In contrast to CXCR2, CXCR1 expression inhibited PSA expression (Figure 2A,C) suggesting that the two receptors may couple to distinct pathways to modulate prostate tumorigenesis.

Another key factor in PCa progression and metastasis is the epithelial–mesenchymal transition (EMT), characterized by the decreased expression of epithelial cell markers, such as E-cadherin and occludins, and the increased expression of mesenchymal markers, such as vimentin and N-cadherin [41,42,43,44,45,46]. CXCR2 expression was previously shown to correlate with EMT and the reorganization of the tumor microenvironment in both C4-2B and LNCaP cells [23]. However, the results herein demonstrated that, despite inhibiting tumor growth, CXCR1 was more effective in regulating EMT marker expression than CXCR2. Vimentin expression was upregulated in MDA-PCa-2b-CXCR1 cells (~3-fold over basal; Figure 2D,G), as compared to CXCR2-expressing cells (1.2-fold over basal). In addition, MDA-PCa-2b-CXCR1 cells displayed a spindle-like phenotype; whereas, MDA-PCa-2b-CXCR2 cells showed a more cuboidal-like phenotype (Figure 1C). The reason for this discrepancy remains unclear. However, cell migration in response to CXCL8 (Figure 1D) was slower in MDA-PCa-2b-CXCR1 cells relative to MDA-PCa-2b-CXCR2 cells.

A third factor that could account for the differential effect of CXCR1 versus CXCR2 expression in prostate tumorigenesis could be the expression of chemokines in the tumor microenvironment (TME). Tumors from MDA-PCa-2b-CXCR1 exhibited a marked increase in the intratumoral expression of murine chemokines relative to MDA-PCa-2b-CXCR2 and control cells (Figure 6, Appendix A). Several of these chemokines including CCL21 (~224-fold) and CXCL13 (~66-fold) were previously shown to exert immunosuppressive effects by recruiting CD4 and CD8 T cells and DC in the TME, thereby inhibiting tumor progression and metastasis [47]. In the well-characterized TRAMP mouse model of PCa, the direct expression of CCL21 in the prostate TME increased CD8 T cells and DC, thereby inhibiting tumor growth and metastasis [48]. Both SCID mice and nude mice lack T cells [49,50]. In a SCID model of lung cancer, however, the direct injection of CCL21 into the TME inhibited angiogenesis and, thereby, tumor growth and metastasis [51]. This process was independent of T cells and was mediated via the activation of CXCR3 receptors by CCL21 [51]. In a nude mice model of colon carcinoma, Vicari et al. [52] also showed that the antitumor effect of CCL21 was mediated by both CXCR3-induced angiostasis and leukocytes recruitment to the TME. The overexpression of chemokines in a nude mice model of CHO was also shown to inhibit tumor growth via the recruitment and activation of neutrophilic granulocytes at the TME [53]. Therefore, it could be that the immunosuppressive effect of CXCR1 expression in tumor growth is due to both the activation of pro-angiostatic chemokine receptors as well as the recruitment and activation of leukocytes into the tumor microenvironment. Indeed, CXCR1-expressing tumors displayed a greater accumulation of NK cells (~5-fold) in the TME, as compared to CXCR2-expressing tumors (~2.5-fold; Figure 5E). Furthermore, VEGF expression showed a ~40% decrease in CXCR1-expressing tumors (Figure 5F). However, MDA-PCa-2b-CXCR1 xenografts in NOD scid gamma (NSG) mice, which lack NK cells (Figure 5G–I), showed no difference in tumor development relative to nude mice (Figure 3B–D). These data suggest that NK accumulation in the tumor microenvironment is not a factor in the suppressive effect of CXCR1 expression in MDA-PCa-2b tumorigenesis.

RNA sequencing analysis revealed distinct patterns of gene regulation between MDA-PCa-2b-CXCR1 and control MDA-PCa-2b-Vec cells (Figure 6). The significant upregulation of genes, including two tumor suppressors (ITM2A and TUSC3), were observed (Figure 6C). Western blot analysis of cell lysates, however, confirmed the overexpression of ITM2A in MDA-PCa-2b-CXCR1 cells (Figure 6E). ITM2A is a type II transmembrane protein that possesses the conserved BRICHOS domain that is characteristic of the BRICHOS superfamily of proteins. ITM2A was previously shown to regulate T cell development as well as the differentiation of various cell types, including bone, epithelial, and mesenchyma stem cells [54,55]. Recent studies, however, have highlighted ITM2A tumor-suppressive effects in various cancer types, including breast and ovarian cancers [56,57,58]. Our attempt to inhibit ITM2A expression in MDA-PCa-2b-CXCR1 cells was unsuccessful. However, the overexpression of ITM2A in MDA-PCa-2b cells decreased tumor growth similar to that of MDA-PCa-2b-CXCR1 cells. These data suggest that the suppressive effect of CXCR1 in MDA-PCa-2b is likely due to the upregulation of ITM2A expression. Supporting that contention is that ITM2A gene transcripts were also shown to be downregulated in human prostate adenocarcinoma patients compared to control subjects (Figure 6F).

## 5. Conclusions

In summary, the data herein demonstrate that the IL-8/CXCL8 receptor’s expression in prostate TME can be both beneficial and harmful. CXCR1 inhibits; whereas, CXCR2 promotes tumor development. The tumor-suppressive effect of CXCR1 is unclear but seems to correlate with an increase expression of the tumor suppressor ITM2A. Further studies are needed to fully understand the antitumor activity of CXCR1 in prostate tumorigenesis.

## Figures and Tables

**Figure 1 cancers-16-04138-f001:**
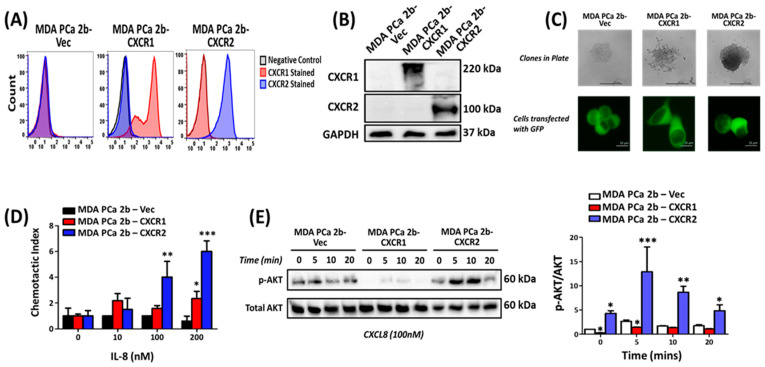
Expression and characterization of MDA-PCa-2b cells stably expressing CXCR1 and CXCR2. (**A**) MDA-PCa-2b cells were transfected with pcDNA3.1-C-(k)DYK plasmid containing CXCR1 (MDA-PCa-2b-CXCR1) or CXCR2 (MDA-PCa-2b-CXCR2) constructs or vector alone (MDA-PCa-2b-Vec). G418-resistant cells were sorted using receptor-specific antibodies. Depicted are representative FACS analyses of at least 5 experiments. (**B**) Western blot analysis of MDA-PCa-2b cells expressing empty vector (lane 1), vector containing CXCR1 (lane 2), or CXCR2 (lane 3). (**C**) Representative image of a single clone morphology following clonogenic assay and transfection of cells with GFP protein. (**D**) For the chemotaxis assay, the dose response of IL-8-induced cell migration was assessed using the NeuroProbe chemotaxis plate. Graphical quantification of the chemotaxis index at 0, 10, 100, and 200 nM of IL-8 are shown. Results are representative of two independent experiments performed in quadruplets. (**E**) Representative Western blotting image and graphical quantification of average band density for phospho-AKT and total AKT following IL-8 induced AKT phosphorylation. * *p* < 0.05, ** *p* < 0.01, and *** *p* < 0.001.

**Figure 2 cancers-16-04138-f002:**
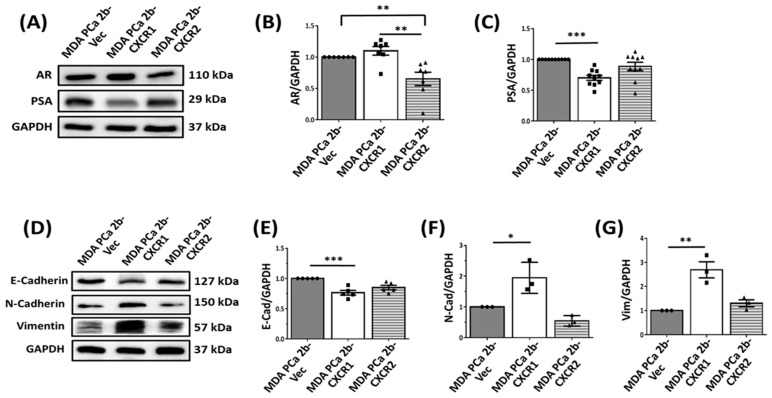
Effects of CXCR1 and CXCR2 overexpression on AR, PSA, and EMT marker expression in MDA-PCa-2b cells. (**A**) MDA-PCa-2b-Vec, MDA-PCa-2b-CXCR1, and MDA-PCa-2b-CXCR2 cell lysates were assayed by Western blotting for AR, PSA, and GAPDH using specific antibodies. (**B**,**C**) The graphical quantification of band density analysis for AR (**B**) and PSA (**C**), relative to GAPDH. (**D**) Cell lysates were assayed by Western blotting for E-cadherin, N-cadherin, and vimentin and GAPDH expression using specific antibodies. Graphical quantification for E-cadherin (**E**), N-cadherin (**F**), and vimentin (**G**) expression, relative to GAPDH. Data were obtained from at least three independent experiments. * *p* < 0.05, ** *p* < 0.01, and *** *p* < 0.001.

**Figure 3 cancers-16-04138-f003:**
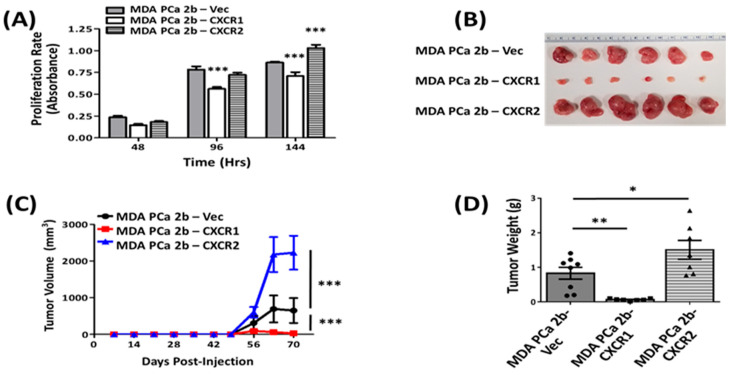
Effect of CXCR1 and CXCR2 overexpression on in-vitro and in-vivo growth of MDA-PCa-2b cells. (**A**) In vitro cell proliferation rates for MDA-PCa-2b-CXCR1, MDA-PCa-2b-CXCR2, or MDA-PCa-2b-Vec cells were determined using the MTT assay, as described in Section 2. Results are expressed as absorbance at 570 nm. The data are representative of one of three independent experiments. (**B**–**D**) For tumor xenografts, cells (5 × 10^6^ cells) were injected subcutaneously into 6–8-week-old nude mice. Tumor growth was monitored weekly until the mice were euthanized, and the tumor weight was determined as described in Section 2. Shown are the representative images of harvested tumors (**B**), tumor volume measured over time (**C**), and the tumor weight of mice [n = at least 7 mice per group] (**D**). Results shown are representative of three independent experiments. * *p* < 0.05, ** *p* < 0.01, and *** *p* < 0.001.

**Figure 4 cancers-16-04138-f004:**
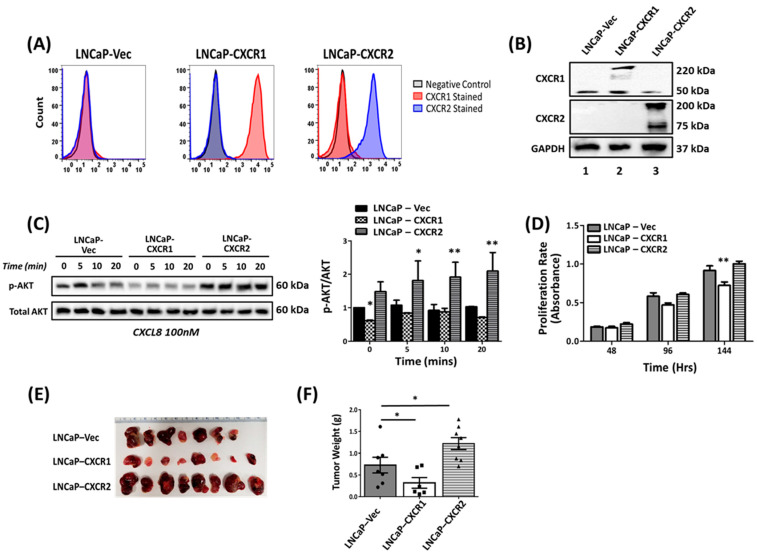
Effect of CXCR1 and CXCR2 overexpression in LNCaP cells. LNCaP cells were transfected with pcDNA3.1-C-(k)DYK plasmid containing CXCR1 (LNCaP-CXCR1), CXCR2 (LNCaP-CXCR2), or vector alone (LNCaP-Vec). G418-resistant cells were sorted using receptor-specific antibodies. (**A**) Representative FACS analysis of 3 experiments. (**B**) Western blot analysis of LNCaP-Vec (lane 1), LNCaP-CXCR1 (lane 2), and LNCaP-CXCR2 (lane 3). (**C**) Representative Western blotting image and graphical quantification of average band density for phospho-AKT and total AKT following IL-8-induced AKT phosphorylation. (**D**) In vitro cell proliferation rates for LNCaP-CXCR1, LNCaP-CXCR2, and control LNCaP-Vec cells were determined using the MTT assay. The data shown are representative of two independent experiments. (**E**,**F**) For tumor xenografts, cells (5 × 10^6^ cells) were injected subcutaneously into 6–8-week-old nude mice [n = 8 mice per group], and animals were monitored weekly. Animals were euthanized, and the tumor weight normalized by the weight of the mice was determined. Shown are representative tumors (**E**) and the average tumor weight (**F**) of three independent experiments. * *p* < 0.05, ** *p* < 0.01.

**Figure 5 cancers-16-04138-f005:**
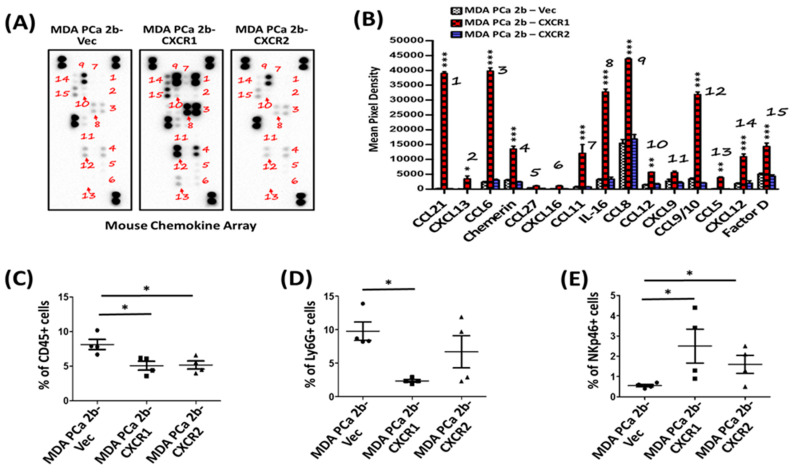
Analysis of chemokines and VEGF expression and immune cell infiltration in MDA-PCa-2b-CXCR1 and MDA-PCa-2b-CXCR2 tumor microenvironments. (**A**,**B**) Tumor lysates from MDA-PCa-2b-Vec, MDA-PCa-2b-CXCR1, and MDA-PCa-2b-CXCR2 xenografts were assayed for a variety of chemokines using the mouse chemokine array as described in Section 2. (**A**) Representative images and (**B**) the graphical quantification of chemokine band densities. The experiment was repeated twice. * *p* < 0.05, ** *p* < 0.01, and *** *p* < 0.001. (**C**–**E**) Single-cell isolates from MDA-PCa-2b-Vec, MDA-PCa-2b-CXCR1, and MDA-PCa-2b-CXCR2 tumor xenografts were stained for different leukocyte subpopulations (CD45, LY6G, and NKp46) and were analyzed by a FACScan flow cytometer using CellQuest software. Graphical plots for CD45 (**C**), LY6G (**D**), and NKp46 (**E**) are shown. * *p* < 0.05. (**F**) Lysates from different tumors were assayed by Western blot for VEGF expression. Representative Western blot image and graphical representation of band densities for VEGF relative to GAPDH are shown. (**G**–**I**) For tumor xenografts, cells (5 × 10^6^ cells) were injected subcutaneously into 6–8-week-old NSG mice [n = 5 mice per group]. Tumor growth was monitored weekly until mice were euthanized, and the tumor weight was determined as described in Section 2. Shown are the representative images of harvested tumors (**G**), tumor volume measured over time (**H**), and the tumor weight (g) of mice (**I**). * *p* < 0.05, ** *p* < 0.01, and *** *p* < 0.001.

**Figure 6 cancers-16-04138-f006:**
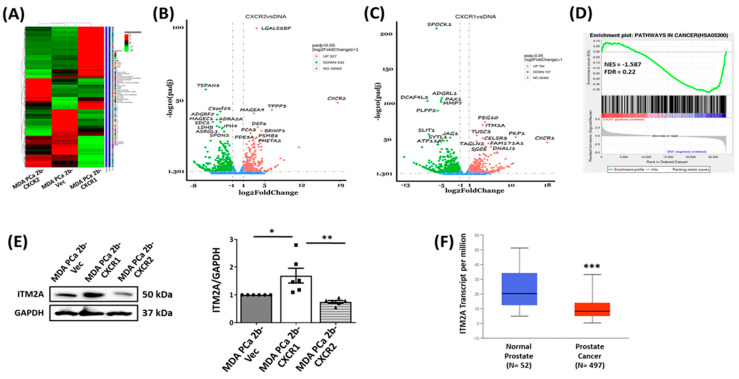
Overexpression of CXCR1 upregulates ITM2A expression. (**A**) Heat map showing the differential expression of genes in MDA-PCa-2b-Vec, MDA-PCa-2b-CXCR1, and MDA-PCa-2b-CXCR2. (**B**) Volcano plot showing the significantly expressed genes in MDA-PCa-2b-CXCR2 versus MDA-PCa-2b-Vec cells. (**C**) Volcano plot showing the significantly expressed genes in MDA-PCa-2b-CXCR1 versus MDA-PCa-2b-Vec cells. (**D**) GSEA analysis showing the downregulation of genes associated with cancer pathways in MDA-PCa-2b-CXCR1, when compared to MDA-PCa-2b-Vec cells. (**E**) Cell lysates were assayed by Western blot for ITM2A expression. Representative Western blot image and the graphical representation of band densities for ITM2A relative to GAPDH are shown. (**F**) ITM2A gene expression analysis of TCGA data between human prostate cancer and normal prostate tissues, extracted from UALCAN. Data are depicted as ITM2A transcript number per million. * *p* < 0.05, ** *p* < 0.01, and *** *p* < 0.001.

**Figure 7 cancers-16-04138-f007:**
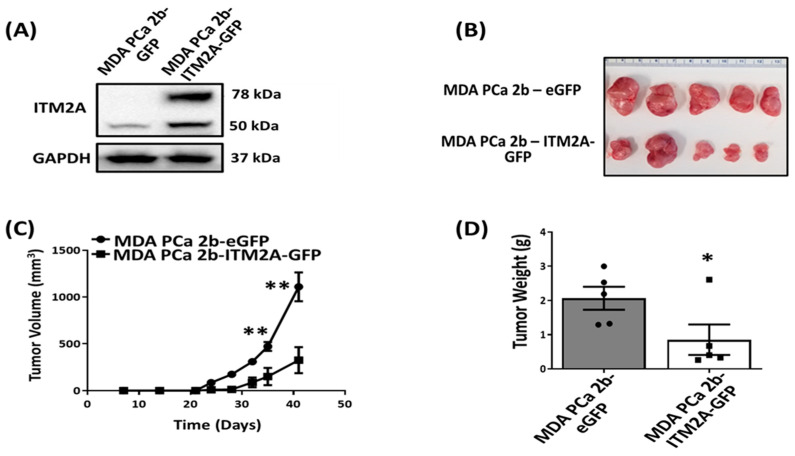
Effect of ITM2A overexpression on growth of MDA-PCa-2b cells. (**A**) Cells were transfected with pcDNA3.1-C-eGFP plasmid containing either human ITM2A or control vector, and G418-resistant clones (MDA-PCa-2b-ITM2A-GFP or MDA-PCa-2b-eGFP) were obtained. (**A**) Western blot analysis validating the overexpression of ITM2A in MDA-PCa-2b cells. (**B**) Representative tumors were harvested from nude mice, following the subcutaneous injection of MDA-PCa-2b-eGFP and MDA-PCa-2b-ITM2A-GFP cells [n = 5 mice per group] (**C**) The tumor volume was monitored and measured over the course of 8 weeks. (**D**) Graphical quantification of the average weight (g) of harvested tumors for MDA-PCa-2b-eGFP and MDA-PCa-2b-ITM2A-GFP cells. * *p* < 0.05 and ** *p* < 0.01.

## Data Availability

The original contributions presented in this study are included in the article. Further inquiries can be directed to the corresponding author.

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
