# Peer review of "CXCR1 Expression in MDA-PCa-2b Cell Upregulates ITM2A to Inhibit Tumor Growth"

_cancers, 2024, doi:10.3390/cancers16244138_

Round 1

Reviewer 1 Report

Comments and Suggestions for Authors

In the article titled "CXCR1 Expression in MDA-PCa-2b Cells Upregulates ITM2A to Inhibit Tumor Growth," the authors stably expressed CXCR1 in the androgen-dependent MDA-PCa-2b cell line to investigate the role of CXCR1 prostate cancer. They also utilized cell-line-derived xenografts and conducted a variety of molecular and functional analyses. The authors conclude that CXCR1 expression in MDA-PCa-2b cells upregulate the tumor supressor ITM2A, leading to tumor inhibition. This manuscript provides a thorough study of the impact of CXCR1 overexpression in MDA-PCa-2b cells, with some additional experiments performed in LNCaP cells . The methodology is clearly described, and most conclusions are well-supported by the presented data. However, there are a few points the authors should address:

  • Simple Summary, line 16: “CXCR1 inhibits via ITM2A upregulation” — specify what process or outcome is inhibited by CXCR1 through ITM2A upregulation.
  • Methods: Specify that NOD scid gamma (NSG) mice were also obtained from Jackson Laboratory, alongside “Nude (Nu/J) mice” from the same source. Also, standardize nomenclature for “male athymic nude” mice across the manuscript.
  • In the clonogenic experiment (Figure 1C), the number of colonies per condition is missing. Enlarging images is suggested for clearer assessment.
  • GFP transfection is not described in the Methods section, nor is fluorescence imaging. Please include these details.
  • The clonogenic figure legend needs further explanation of the magnification in each picture. Explain why fluorescent images show only a few cells outside colonies. If morphological differences are relevant, consider quantifying and statistically analyzing them to support conclusions (lines 238-240).
  • Include the number of replicates and animals for each experiment.
  • “CXCL8 induced time-dependent phosphorylation of AKT in both MDA-PCa-2b-CXCR1 and MDA-PCa-2b-CXCR2 (Fig. 1E). However, AKT phosphorylation in MDA-PCa-2b-CXCR1 was markedly downregulated compared to MDA-PCa-2b-CXCR2 and control cells (Fig. 1E).” Include appropriate statistical analysis for time-point comparisons to support claims of time-dependent phosphorylation.
  • “MDA-PCa-2b-CXCR2 cells (Fig. 2A, upper panel) showed a significant decrease in AR expression (0.651 ± 0.107, Fig. 2B) relative to MDA-PCa-2b-CXCR1 (1.099 ± 0.069) and MDA-PCa-2b-Vec (1.0 ± 0.013) cells (Fig. 2B).” Confirm if this decrease is statistically significant for MDA-PCa-2b-CXCR2 vs MDA-PCa-2b-CXCR1.
  • Validation: Key assays (e.g., clonogenic, AKT phosphorylation, AR, PSA, E-cadherin, N-cadherin, Vimentin) should be performed in LNCaP-CXCR1 as well, as stated in the journal guidelines.
  • Sections 3.4, 3.5, and 3.6 should include validation in LNCaP-transfected cells or a reference if previously published.
  • Immune system assessment: Provide a brief discussion on how immunodeficient mice may affect study conclusions.
  • RNA-seq Analysis: The current approach appears gene-biased; a broader analysis would increase the quality of the manuscript. Include ITM2A expression in LNCaP to validate findings.
  • Sections 3.5 and 3.6: Additional evidence is needed to substantiate the ITM2A-mediated CXCR1 anti-tumor mechanism. The experimental design in Section 3.6 does not clearly link ITM2A with CXCR1; more rigorous validation is recommended.
  • Terminology: Unify "PCa" or "prostate cancer" throughout the text.
  • Figure Legends: Indicate the number of replicates for each experiment. Enlarge figure panels for readability.
  • Spacing Issues: Ensure consistent spacing, e.g., Lines 111, 155, 219, 403, 430, 456.
  • Please separate multiple affiliations with commas.
  • Figures: Ensure uniform font across figures (e.g., Fig. 4A and B).
  • Supplementary Figures: S2 is cited before S1. Rearrange to reflect the correct sequence.

Author Response

Reviewer 1

We thank the reviewer for the positive comments and constructive suggestions of the manuscript. Below are point-to-point responses to the critiques:

  • Simple Summary, line 16: “CXCR1 inhibits via ITM2A upregulation” — specify what process or outcome is inhibited by CXCR1 through ITM2A upregulation.

The sentence has been revised to indicate that  “CXCR1 expression, not CXCR2, upregulates the tumor suppressor ITM2A to inhibit tumor growth”.

  • Methods: Specify that NOD scid gamma (NSG) mice were also obtained from Jackson Laboratory, alongside “Nude (Nu/J) mice” from the same source. Also, standardize nomenclature for “male athymic nude” mice across the manuscript.

The methods section (2.10) has been revised to include the source of the animals (line 187 -189)

  • In the clonogenic experiment (Figure 1C), the number of colonies per condition is missing. Enlarging images is suggested for clearer assessment.

We thank the reviewer for pointing that out. The images displayed in figure 1C, top panels, are from single clones. To clarify this point, the figure legend for “Figure 1C” has been revised. It now states, “Representative image of a single clone morphology…” (line 280 -281)

  • GFP transfection is not described in the Methods section, nor is fluorescence imaging. Please include these details.

We have included the details in the Materials & Methods section 2.3 (lines 121 – 125)

  • The clonogenic figure legend needs further explanation of the magnification in each picture. Explain why fluorescent images show only a few cells outside colonies. If morphological differences are relevant, consider quantifying and statistically analyzing them to support conclusions (lines 238-240).

The magnification of both the clonogenic assay and the fluorescent images are now included the figure panels. We have provided better-quality image of all figures as TIFF images. The clones in plate images are from single clone captured using the Moticam imaging microscope whereas the fluorescent images were captured field view using the Keyence fluorescence microscope. The magnification bars are included in the figures. The Material & Methods section has been amended to clarify these points.  

  • Include the number of replicates and animals for each experiment.

The number of independent repetitions for the animal studies are now included in the figure legends.

  • “CXCL8 induced time-dependent phosphorylation of AKT in both MDA-PCa-2b-CXCR1 and MDA-PCa-2b-CXCR2 (Fig. 1E). However, AKT phosphorylation in MDA-PCa-2b-CXCR1 was markedly downregulated compared to MDA-PCa-2b-CXCR2 and control cells (Fig. 1E).” Include appropriate statistical analysis for time-point comparisons to support claims of time-dependent phosphorylation.

Figure 1E includes the level of significance for each time point (0, 5, 10, 20) and compares AKT phosphorylation levels between receptor overexpressing cells and control cells expressing the vector alone.

  • “MDA-PCa-2b-CXCR2 cells (Fig. 2A, upper panel) showed a significant decrease in AR expression (0.651 ± 0.107, Fig. 2B) relative to MDA-PCa-2b-CXCR1 (1.099 ± 0.069) and MDA-PCa-2b-Vec (1.0 ± 0.013) cells (Fig. 2B).” Confirm if this decrease is statistically significant for MDA-PCa-2b-CXCR2 vs MDA-PCa-2b-CXCR1.

The level of AR expression was significantly decreased in MDA-PCa-2b-CXCR1, relative to both MDA-PCa-2b-CXCR2 and MDA-PCa-2b-Vec cells. We have modified the figure 2B to clearly reflect the statistical analysis between MDA-PCa-2b-CXCR2 and MDA-PCa-2b-CXCR1.

  • Validation: Key assays (e.g., clonogenic, AKT phosphorylation, AR, PSA, E-cadherin, N-cadherin, Vimentin) should be performed in LNCaP-CXCR1 as well, as stated in the journal guidelines.

As indicated in manuscript, the experiments were also conducted with LNCaP cells stably expressing CXCR1 and CXCR2. Since most of the information in AKT, AR, PSA, E-cadherin, N-cadherin, Vimentin were previously reported by Li et al (2019) [ref. # 23]), we only included in supplementary Figure S2 (and have now added supplementary Figure S3) additional data confirming that in LNCaP cells, CXCR1 expression also inhibited tumor growth.

  • Sections 3.4, 3.5, and 3.6 should include validation in LNCaP-transfected cells or a reference if previously published.

We conducted similar experiments in LNCaP cells. As stated above, some information about CXCR2 overexpression in LNCaP have been previously reported (Li et al (2019) [ref. # 23]). We only included in supplementary Fig S2 additional data confirming the opposing effect of CXCR1 and CXCR2 expression on tumor growth.

  • Immune system assessment: Provide a brief discussion on how immunodeficient mice may affect study conclusions.

Our analysis of murine chemokines expression in the tumor microenvironment (TME) has revealed a significant increase in chemokine expression (Fig. 4B). However, our analysis of immune cells infiltrate in the TME showed only a slight increase in Natural Killer (NK) cells accumulation in CXCR1 relative to CXCR2 tumors.  Thus, to rule out the influence of NK cells on the decreased tumorigenesis observed in MDA-PCa-2b-CXCR1, we carried out tumor xenografts in NSG mice which lacks B, T and NK cells, and the results were similar to that of nude mice (Fig 4 G-I).  This information is documented in both in the results (lines 411 – 419) and discussion (lines 512 – 515) sections.

  • RNA-seq Analysis: The current approach appears gene-biased; a broader analysis would increase the quality of the manuscript. Include ITM2A expression in LNCaP to validate findings.

RNA seq analysis of LNCaP-CXCR2 and LNCaP-DNA were previously reported by Li et al (2018) [ref.# 23]. Because of the high cost of RNA seq, we decided to focus our RNA-seq Analysis on generating new information from the MDA-PCa-2b cell lines.

  • Sections 3.5 and 3.6: Additional evidence is needed to substantiate the ITM2A-mediated CXCR1 anti-tumor mechanism. The experimental design in Section 3.6 does not clearly link ITM2A with CXCR1; more rigorous validation is recommended.

We appreciate this comment by the reviewer and agree that there is indeed more to be known of ITM2A-mediated CXCR1 anti-tumor mechanism. Our results thus far from overexpressing ITM2A indicates an association. As indicated in the result and discussion sections, we attempted multiple gene editing approaches (including CRISPR knockout, siRNA from different sources) to suppress ITM2A expression in the MDA-PCa-2b-CXCR1 cells and other generated cells but were unsuccessful (line 450 – 455; 525 – 529).  We are currently exploring new approaches and collaborating with 2 groups to establish stable MDA-PCa-2B cell lines deficient in ITMA2 expression.  Our future studies aim to fully decipher the CXCR1-ITM2A axis in prostate tumorigenesis.

  • Terminology: Unify "PCa" or "prostate cancer" throughout the text.

We thank reviewer for this suggestion and have revised accordingly.

  • Figure Legends: Indicate the number of replicates for each experiment. Enlarge figure panels for readability.

We have provided TIFF format of all figures for clarity.

  • Spacing Issues: Ensure consistent spacing, e.g., Lines 111, 155, 219, 403, 430, 456.

We have revised spacing issues, as suggested.

  • Please separate multiple affiliations with commas.

We have revised as suggested.

  • Figures: Ensure uniform font across figures (e.g., Fig. 4A and B).

We have revised as suggested.

  • Supplementary Figures: S2 is cited before S1. Rearrange to reflect the correct sequence.

We have revised as suggested.

Reviewer 2 Report

Comments and Suggestions for Authors

I applaud the authors on tackling the question of CXCR1's role in prostate cancer tumorigenesis in light of the well-described role of CXCR2.  Specific comments are as follows:

Abstract/Introduction:

-Line 51-52: Would change ADT to AR-targeted therapies since we now routinely treat hormone sensitive disease with intensified regimens of ADT plus AR pathway inhibitor

-Line 53: Not all CRPC is "aggressive", some can still have indolent clinical course such as with PSA only progression

-Line 54: While it is true that various androgen-independent pathways can occur, it should be noted that AR signaling is often still utilized in CRPC with various resistance mechanisms such as amplification, alternative ligands, different coactivators, etc.

-Line 61-62: Mention of CXCR2 expression in cancer cells but is it only cancers or is it expressed in the normal epithelial tissues for these cancers as well?

-Lines 68-74: There is mention of cell line used but no mention of the animal model used, may be good to introduce in vivo studies here as well

Methods: 

-Line 178: Would clarify the cells here like you did in lines 284-285 of results section

-2.10: In addition to the 5-10 week time frame, would clarify if a maximum tumor volume was used as threshold as well that triggered humane mouse euthanasia 

Results:

-Line 252: You mention it is described with CXCR2, but did you assess AR, PSA, and EMT marker change with CXCR1 overexpression in LNCaP as well?  

-3.4 title is awkwardly worded, recommend editing

Discussion:

-Line 389: When discussing "androgen-independent", are we referring to cells with intact AR signaling that no longer requires androgen as ligand or cells that no longer utilize AR signaling all together (such as NE prostate cancer)?  Important distinction to make.

-Line 394: Why do we think there was discrepancy with decrease in AR expression with CXCR2 but PSA expression decrease was with CXCR1?  One would expect these to track together since PSA is end result of AR signaling 

-Line 407: Perhaps marker expression as defined here with E-cadherin, N-cadherin, and vimentin expression is not a sufficient definition to enrich for EMT.  May be worth exploring alternative NE definitions such as previously used gene expression signatures?

Conclusion:

-455: Do you mean tumor suppressive effect?

Author Response

Reviewer 2

We appreciate the reviewer’s positive comments regarding the manuscript and for applauding our effort on tackling the role of CXCR1 in prostate cancer tumorigenesis. Below are point-to-point responses to the criticisms and suggestions

 Abstract/Introduction:

-Line 51-52: Would change ADT to AR-targeted therapies since we now routinely treat hormone sensitive disease with intensified regimens of ADT plus AR pathway inhibitor

We thank the reviewer for this suggestion and have revised the manuscript accordingly.

-Line 53: Not all CRPC is "aggressive", some can still have indolent clinical course such as with PSA only progression

We have edited the sentence. It now reads as “…an often  aggressive metastatic form of PCa”.

-Line 54: While it is true that various androgen-independent pathways can occur, it should be noted that AR signaling is often still utilized in CRPC with various resistance mechanisms such as amplification, alternative ligands, different coactivators, etc.

We totally agree with this assertion and input by the reviewer.

-Line 61-62: Mention of CXCR2 expression in cancer cells but is it only cancers or is it expressed in the normal epithelial tissues for these cancers as well?

We thank reviewer for this comment. It has been reported in several studies that malignant cells often have upregulated levels of CXCR2, when compared to normal epithelial cells.

-Lines 68-74: There is mention of cell line used but no mention of the animal model used, may be good to introduce in vivo studies here as well

We thank reviewer for bringing this to our attention. As indicated in Materials and Methods (section 2.10. [lines 187 – 189]) these are well-established and commercially available mouse models

Methods: 

-Line 178: Would clarify the cells here like you did in lines 284-285 of results section

We have revised section as suggested by reviewer

-2.10: In addition to the 5-10 week time frame, would clarify if a maximum tumor volume was used as threshold as well that triggered humane mouse euthanasia.

Thank you for this great input. From our experience and publications over several years using these cell lines, within this time frame, they do not exceed the tumor-animal weight ratio stipulated in our IACUC protocol and mice are monitored regularly for any tumor related distress or burden. As such our manuscript states the 5-10 weeks.

Results:

-Line 252: You mention it is described with CXCR2, but did you assess AR, PSA, and EMT marker change with CXCR1 overexpression in LNCaP as well?  

We thank reviewer for this point. Differential expression of this proteins in CXCR1, CXCR2 LNCaP cells were assessed in LNCaP cells. The data were not in the earlier submission but have now been included in the supplementary data section of the revised submission. (Supplementary Figure S3).

-3.4 title is awkwardly worded, recommend editing

We thank reviewer for this recommendation.

Discussion:

-Line 389: When discussing "androgen-independent", are we referring to cells with intact AR signaling that no longer requires androgen as ligand or cells that no longer utilize AR signaling all together (such as NE prostate cancer)?  Important distinction to make.

The sentence between line 388 -290 was in reference to other studies (ref # 35, 39, 40) that earlier reported that the chemokine ligand, CXCL8, promoted androgen-independent prostate tumorigenesis.

-Line 394: Why do we think there was discrepancy with decrease in AR expression with CXCR2 but PSA expression decrease was with CXCR1?  One would expect these to track together since PSA is end result of AR signaling.

Several studies have reported on the non-canonical AR signaling axis whose effects may or may not directly impact PSA expression levels (refs: 7, 8, 11). In our study, CXCR2 overexpression in MDA-PCa-2b lowered AR protein levels, albeit insignificantly.

-Line 407: Perhaps marker expression as defined here with E-cadherin, N-cadherin, and vimentin expression is not a sufficient definition to enrich for EMT.  May be worth exploring alternative NE definitions such as previously used gene expression signatures?

We agree with reviewer that there may be other underlying processes responsible for the decreased migration of MDA-PCa-2b-CXCR1 following CXCL8 stimulation other than their EMT characteristics. The findings from our gene expression signature however indicated highly positive Mesenchymal phenotypic signatures in the MDA-PCa-2b-CXCR1 cells and this is also validated by their morphological phenotype.

Conclusion:

-455: Do you mean tumor suppressive effect?

We thank the reviewer for this comment. We meant “tumor suppressive” and have revised the manuscript accordingly.

Reviewer 3 Report

Comments and Suggestions for Authors

In this study, Adekoya et al. discovered that CXCR1 but not CXCR2 expression induces ITM2A expression and reduce cell proliferation / tumor growth. The entire study is mainly based on observations from MDA-PCa-2b cells, which is not a conventional type of prostate cancer cell line, i.e., commonly used by prostate cancer researchers for basic science-type of studies. Indeed, MDA-PCa-2b cells are from a bone metastasis of androgen-independent adenocarcinoma of the prostate, although it does express androgen receptor (AR). There is a lack of evidence from other typical AR-dependent prostate cancer cell lines, such as LNCaP or its derivatives, as well as cells containing AR variants (e.g., 22RV1, CWR-R1-D567), and AR-null cells (e.g., PC3, DU145). More importantly, the role of CXCR1 in androgen-independent prostate cancer has already been studied in a 2009 Cancer Research paper (Shamaladevi N et al., CXC receptor-1 silencing inhibits androgen-independent prostate cancer, Cancer Res, 2009). Therefore the relevance and the novelty of the findings from this paper can be limited.

The main novel finding from this paper is the discovery that CXCR1 induces the expression of ITM2A, which is a tumor suppressor, overexpression of ITM2A reduced MDA-PCa-2b tumour growth. However, the key point should be to prove that the induction of ITM2A expression is crucial for the anti-prostate tumour effect of CXCR1, so the most important experiment should be to knock down or to genetically delete ITM2A in MDA-PCa-2b-CXCR1 cells and see whether the absence of ITM2A rescues the growth inhibitory effect of CXCR1 overexpression on MDA-PCa-2b-CXCR1 cells.

Author Response

Reviewer 3

 We sincerely appreciate the reviewer’s comments and objective assessment of our manuscript. Below are our responses to the respective concerns that were raised:

  • “The entire study is mainly based on observations from MDA-PCa-2b cells, which is not a conventional type of prostate cancer cell line, i.e., commonly used by prostate cancer researchers for basic science-type of studies”

 In addition to MDA-PCa-2b we also used LNCaP transfected cells to demonstrate that CXCR1 expression also inhibited tumor growth whereas CXCR2 expression promoted tumor growth (Fig S1). Since the data on LNCaP-CXCR2 were previously reported (ref #23) we focused our attention on MDA-PCa-2b.  It is also worth mentioning that the MDA-PCa-2b is the only validated African American derived prostate cancer cell line which has been used extensively by biomedical researchers conducting both health-disparity and non-health disparity research.

  • There is a lack of evidence from other typical AR-dependent prostate cancer cell lines, such as LNCaP or its derivatives, as well as cells containing AR variants (e.g., 22RV1, CWR-R1-D567), and AR-null cells (e.g., PC3, DU145).

We thank the reviewer for this comment. As we indicated above, our study utilized both MDA-PCa-2b and LNCaP cells. Our goal in this manuscript was to focus on the AR-dependent prostate tumorigenesis. As such, we did not utilize AR-null cells. In addition, the role of CXCR1 in androgen-independent prostate cancer has already been studied by Shamaladevi N et al., CXC receptor-1 silencing inhibits androgen-independent prostate cancer, Cancer Res, 2009.

  • More importantly, the role of CXCR1 in androgen-independent prostate cancer has already been studied in a 2009 Cancer Research paper (Shamaladevi N et al., CXC receptor-1 silencing inhibits androgen-independent prostate cancer, Cancer Res, 2009).

We again thank the reviewer for bringing up this earlier study in which CXCR1 was silenced by shRNA in PC3 cell line. Again, Shamaladevi’s paper was focused on androgen-independent prostate cancer. While certain studies have been able to identify the CXCR2-positive cells with neuroendocrine (NE) phenotypes (ref. 20, 23), a fundamental question in prostate cancer biology is what happens to the CXCR1-positive cells as prostate tumorigenesis advances? Our study is the first to comparatively assess the roles of both CXCR1 & CXCR2 synchronously. We believe this new findings sheds understanding to the fate of prostate cells overexpressing CXCR1.

  • …so the most important experiment should be to knock down or to genetically delete ITM2A in MDA-PCa-2b-CXCR1 cells and see whether the absence of ITM2A rescues the growth inhibitory effect of CXCR1 overexpression on MDA-PCa-2b-CXCR1 cells

We certainly agree with the reviewer on this rescue experiment and we made all good-faith attempts (and we continue to do so) at completing ITM2A knock-down and knock-in experiments during the course of our study (see result section, [lines 450 – 455]; and discussion section, [lines 525 – 527]) To reiterate, we utilized multiple CRISPR and siRNA targets from different sources, including Origene, GeneScript, SantaCruz, among others but all attempts at knocking down ITM2A in MDA-PCa-2b-CXCR1 cells were unsuccessful.  We are currently working with 2 collaborators in an effort to identify a better approach to generate MDA-PCa-2b cell lines deficient in ITM2A expression.

Round 2

Reviewer 3 Report

Comments and Suggestions for Authors

The manuscript hasn't been improved compared to the previous submitted version.

Author Response

Comment 1: The manuscript hasn't been improved compared to the previous submitted version.

Response: We appreciate the comment of reviewer. We have made additional revision to the manuscript and provided an additional figure (Figure 4) for the LNCaP cell line containing data from various assays/experiments completed in the MDA PCa 2b cell line, to buttress the findings of our study. As we indicated in our submitted manuscript and earlier comments, our study involved the use of two well characterized androgen-dependent cell lines (MDA PCa 2b & LNCaP cells) and build up on an earlier study on LNCaP-CXCR2  that was previously reported (ref #23) - thus our greater focus on MDA-PCa-2b. Other prostate cancer cell lines are either androgen-independent or derivatives of the LNCaP cell line. We would also wish to highlight that no previous study has comparatively evaluated the tumorigenic effects of CXCR1 & CXCR2 synchronously.

Again, we would also like to mention (and as highlighted in our manuscript both in the result section [lines 434 - 438] and conclusion section [lines 514 - 515]) that we tried multiple approaches at knocking down or knocking out ITM2A in the MDA PCa 2b-CXCR1 cells, using different targets & vectors from multiple sources as well as in collaboration with other labs with additional gene-editing expertise. All these attempts were unsuccessful as the cells after selection do not proliferate; thus our decision to overexpress ITM2A in naive MDA PCa 2b cells.

Asking us to repeat all experiments with an additional cell line(s) (that will be androgen-independent & with entirely different biomolecular characteristic) will certain be unwarranted; with all due respect.

We thank you once again for the review.